# Study on Improving Measures of Mechanical Properties of Geopolymer Materials and Its Effect on CO_2_ Emission

**DOI:** 10.3390/polym15071699

**Published:** 2023-03-29

**Authors:** Jinqian Luo, Xiaoshuang Shi, Qingyuan Wang, Jinxin Dai, Xiang Deng, Yu Xue

**Affiliations:** 1Key Laboratory of Deep Underground Science and Engineering for Ministry of Education, Department of Architecture and Environment, Sichuan University, Chengdu 610065, China; 2Failure Mechanics & Engineering Disaster Prevention and Mitigation, Key Laboratory of Sichuan Province, Sichuan University, Chengdu 610065, China; 3Department of Mechanical Engineering, Chengdu University, Chengdu 610106, China; 4First Construction (Sichuan) Co., Ltd. of China Construction Third Engineering Bureau, Chengdu 610065, China

**Keywords:** geopolymer, CDWC, CaO, compressive strength, CO_2_ emission

## Abstract

Using construction and demolition waste composites (CDWC) and fly ash (FA) to replace cement to produce concrete can reduce CO_2_ emissions. However, the CDWC-based geopolymer materials have two imperfections: the compressive strength is prone to decrease with the increase of curing age (strength shrinkage) under heat curing conditions, and the strength develops slowly under ambient curing conditions. To solve the problems of these materials, on the one hand, we designed an experiment of preparing CDWC-based geopolymer concrete (CDWGC) with pretreated CDWC at different high temperatures. We analyzed the influence of different pretreatment temperatures on the mechanical properties of CDWGC through compressive strength, SEM-EDS and XRD. On the other hand, we added CaO to improve the mechanical properties of CDWC-based geopolymer paste (CDWGP) under ambient curing conditions. Further, the CO_2_ emissions of pretreating CDWC and adding CaO were calculated by life cycle assessment (LCA). The results show that: (1) Pretreatment of CDWC can effectively solve the problem of CDWGC strength shrinkage. (2) The compressive strength of CDWGP cured at ambient can be significantly improved by adding CaO, and the compressive strength can be increased by 180.9% when the optimum content is 3%. (3) Adding CaO had less impact on CO_2_ emissions, a low-carbon way to improve its strength effectively.

## 1. Introduction

Concrete is still the most widely used building material [1], and its demand is still rising [2,3], while cement is the most commonly used binder for traditional concrete. However, producing 1 ton of cement will emit about 1 ton of CO_2_ [4]. Therefore, it is a feasible and necessary measure to seek new low-carbon concrete to replace traditional concrete [5] to reduce CO_2_ emissions in the building materials industry. Geopolymer refers to the precursors rich in aluminosilicate, which takes an alkaline solution as the activator and generates materials with a three-dimensional network structure through geopolymerization under room temperature or low high-temperature conditions [6]. Compared with traditional concrete, geopolymer concrete has better mechanical properties and durability [7,8,9] and can reduce CO_2_ emissions by 62–73% [10,11,12]. Therefore, it is generally believed that geopolymer concrete is a good substitute for traditional concrete [13,14].

With the development of urbanization, China consumes a large number of mineral resources for civil engineering, while construction demolition waste is also increasing. In 2021, China produced 3.209 billion tons of construction waste [15], an increase of 5.63% compared with 2020. However, the recycling rate of construction waste in China is very low, and land landfill is still the main disposal method [16]. Therefore, using construction waste to prepare recycled aggregate, CDWC, and other recycled products is an optimal resource utilization mode. In addition, the preparation of CDWC avoids classifying construction waste. It uniformly grinds the mixture of construction waste (concrete, brick, tile) into powder, which is low in cost and simple in production [17]. What’s more, with the attention paid to the resource value of industrial by-products such as fly ash and slag, the market price is getting higher, and their output is limited, which cannot meet the demand when they are widely used in building materials. Therefore, it is necessary to find alternatives to fly ash and slag.

It has been pointed out that [18,19,20] the CDWC has pozzolanic activity and can be used to replace fly ash and slag to prepare geopolymer concrete. However, due to the low content of Al_2_O_3_ and CaO in CDWC, the pozzolanic activity is relatively low, and their strength develops slowly under ambient curing conditions. For this imperfection, scholars generally adopt the following two ways: Firstly, accelerate early strength development through heat curing. Secondly, increase the content of CaO in the matrix, and the strength development rate under ambient curing conditions can be accelerated. But these two methods are imperfect.

Heat curing can make the mechanical properties of CDWGC develop rapidly [21], but the compressive strength usually declines with the increase of curing age. Alexander [18] et al. prepared geopolymer concrete with CDWC. The 7-day compressive strength can reach 37.5 Mpa under 70 °C curing conditions, but the 28-day compressive strength has only 27 Mpa, with a 28% reduction in strength. Dai [17] et al. and Matteo [22] et al. also found that the compressive strength of geopolymer concrete decreased with curing age under heat curing conditions using the CDWC. It can be seen that high-temperature curing will affect the strength development stability of geopolymer concrete, especially after adding CDWC. For CDWGC prepared through heat curing, scholars pay more attention to explaining the mechanism of heat curing for its strength improvement. However, they tend to ignore the imperfection that heat curing increases the instability of its strength growth.

At present, the way to improve the mechanical properties of geopolymer materials under ambient curing conditions is generally to add materials with high CaO content, such as slag [23,24], Portland cement [25] or Ca(OH)_2_ [6], but these methods have corresponding imperfections. For example, previous studies found that adding slag will significantly shorten the setting time of concrete and affect workability. Although the addition of Portland cement can effectively improve the compressive strength of concrete under ambient curing conditions, it goes against the original intention of geopolymer to reduce CO_2_ emissions without using cement at all. However, Ca(OH)_2_ can easily react with CO_2_ in the air, which weakens the durability of concrete. Therefore, few researchers choose CaO as an additive to improve the strength of concrete. On the other hand, the direct addition of CaO can improve the calcium content of the matrix more efficiently, and the low dosage of CaO will not affect the rapid consumption of CDWC in the geopolymer materials. Therefore, it is practical to explore the feasibility of adding CaO to improve the compressive strength of CDWGP prepared under ambient temperature curing conditions.

To sum up, this paper mainly explores two issues: First, solve the problem that CDWGC’s compressive strength decreases with increased curing age under heat curing conditions. Second, investigate the effect of adding CaO on the mechanical properties of CDWGP under ambient curing conditions and the impact of different improvement measures on CO_2_ emissions.

## 2. Materials and Methods

This article provides a list of English abbreviations involved in the article, as shown in Table 1.

### 2.1. Raw Materials

Fine aggregate is river sand, and the average particle size is 0.46 mm. Coarse aggregate is a continuously graded crushed stone with a particle size of 5~20 mm. FA: provided by Tianjin Zhucheng New Material Co., Ltd. (Tianjin, China), Chemical composition is shown in Table 2, and particle size distribution is shown in Figure 1. CDWC: provided by Sichuan Lianluqiang Environmental Protection Technology Co., Ltd. (Chengdu, China), its chemical composition is shown in Table 2, physical properties are shown in Table 3, and particle size distribution is shown in Figure 1. The chemical composition and particle size of FA and CDWC are obtained by X-ray fluorescence spectroscopy (XRF) and Laser particle sizer, respectively, tested in the Analysis and Test Center of Sichuan University. CaO: provided by Kelong Chemical Reagent Factory (Chengdu, China), with a purity of 98%, in powder form.

The alkaline solution is prepared from sodium silicate solution and sodium hydroxide solution. The sodium silicate solution is provided by Foshan Zhongfa Water Glass Factory (Foshan, China). The modulus (Ms) is 3.13 (Ms = SiO_2_/Na_2_O, Na_2_O = 8.83%, SiO_2_ = 27.64%). NaOH is a block solid with a purity of 98%, provided by Kelong Chemical Reagent Factory (Chengdu, China). Sodium hydroxide solution is prepared by dissolving deionized water with NaOH solid; the concentration is 12 mol/L.

### 2.2. Mix Proportion

This experiment is completed based on the previous work of our research group. According to the conclusion of Dai [17] et al., the CDWGC prepared under the condition of NaOH concentration of 12 mol/L, CDWC content of 20%, FA content of 80%, and curing at 80 °C for 24 h had the optimal compressive strength. To solve the problem of CDWGC strength shrinkage under heat curing conditions, the CDWC is pretreated at different temperatures (40 °C, 60 °C, 80 °C, 100 °C) for 24 h. Then, using pretreated CDWC-prepared concrete. The mix proportion is shown in Table 4. The main factors affecting the strength of concrete are the number and strength of gelling formed by the precursors. At the same time, to avoid the impact of aggregate, the experiment of improving the compressive strength of geopolymer materials by adding CaO under ambient curing conditions is selected as geopolymer paste, in which the CaO content is 0%, 1%, 3%, and 5%. The mix proportion is shown in Table 5.

### 2.3. Experimental Project

#### 2.3.1. Preparation of Specimens

The preparation process of geopolymer concrete specimens is as follows: First, the raw materials weighed according to the mix proportion for the experiment are poured into the concrete forced mixer for mixing for about 1 min. After that, the alkaline solution prepared 24 h in advance is added two times. When thoroughly mixed, cast into the 100 mm × 100 mm × 100 mm steel mold. After vibrating and compacting, cover it with plastic film immediately to avoid the influence of water evaporation on strength development. Then, the mold was placed in an 80 °C oven for curing for 24 h, then de-molded. Finally, placed the specimen at ambient temperature for curing until the test age.

The preparation process of geopolymer paste specimens is as follows: First, pour the weighed CDWC, FA, and CaO into the JJ-5 cement mortar mixer for dry mixing for about 2 min to make the dry materials mix evenly. Then slowly pour the alkaline solution prepared 24 h in advance into the mixer, and continue to stir for 3 min. After mixing, pour the fresh paste into the 40 mm × 40 mm × 160 mm steel mold and immediately cover it with plastic film. After pouring, the specimens shall be cured for 24 h at the ambient temperature of 20 °C and relative humidity of (70 ± 5) % for demolding. Finally, the specimen is cured at ambient temperature until the test time. The schematic diagram of specimen preparation for this experiment is shown in Figure 2.

#### 2.3.2. Experimental Test

The compressive strength of concrete and paste are tested according to GB/T50081-2019 and GB/T17671-1999, respectively. The compressive strength of CDWGC is tested on a 200 t electro-hydraulic servo press machine (Changchun testingmachine factory, Changchun, China) using a load control method with a loading rate of 5 KN/s. The compressive strength of CDWGP is tested on a 60 t electro-hydraulic servo press machine (Changchun testingmachine factory, Changchun, China) and uniformly loaded at a loading rate of 2.4 KN/s until it is destroyed. This experiment tests three specimens for each group, and the average value is taken as the compressive strength result. Compressive strength testing is completed in the Structural Laboratory of Sichuan University. SEM-EDS and XRD are tested by the SU3500 electronic scanning microscope (Oxford Instruments, Tokyo, Japan) and EMPYREAN X-ray diffractometer (Panaco Netherlands, Almelo, The Netherlands) of the Sichuan University Analysis and Testing Center. After drying and vacuuming the CDWC sample, conduct SEM analysis first and quantitatively analyze the elemental composition of CDWC after line scanning the SEM test area to obtain EDS results. The mineral phase in CDWC pretreated at 20 °C, 40 °C, 60 °C, 80 °C, and 100 °C is characterized by XRD (2θ = 10–80°), and the specific phase is qualitatively analyzed using Jade software (6.0). CO_2_ emission calculation is based on the life cycle assessment (LCA) theory, using the first full-featured professional LCA software (3.0) in the LCA system with independent intellectual property rights in China-eBalance and its built-in CLCD database to scientifically quantify the CO_2_ emissions in the improvement mode.

## 3. Results and Discussion

### 3.1. Influence of Pretreatment on the Development of Compressive Strength

The development trend of CDWGC compressive strength with curing age after pretreating CDWC at different temperatures is shown in Figure 3. It can be seen that after pretreatment, the compressive strength of CDWGC increases with the curing age, and there is no abnormal phenomenon that the strength decreases with the increase of curing age. At different pretreatment temperatures, the compressive strength increased by 9.2% at the highest, 0.9% at the lowest, and 4.6% on average from 3 d to 7 d. And from 7 d to 28 d, the compressive strength increased by 10.9% at the highest, 4.3% at the lowest, and 7.3% on average. While from 3 d to 28 d, the compressive strength increased by 17.3% at the highest, 7.6% at the lowest, and 12.2% on average. The macro mechanical properties prove that the CDWC has no negative impact on the development of CDWGC compressive strength when they experience high temperatures. The high-temperature pretreatment of CDWC can effectively solve the problem of CDWGC compressive strength shrinkage.

It is worth noting that pretreat CDWC can improve the compressive strength of CDWGC, as shown in Figure 4. The relationship between pretreatment temperature and concrete compressive strength can be obtained: The compressive strength of the experimental groups (the pretreatment temperature is 40–100 °C) are higher than that of the control group (20 °C). Moreover, in the range of 20–60 °C, the compressive strength increases with the increase of pretreatment temperature, while in 60–100 °C, the compressive strength decreases with the rise in pretreatment temperature. Although the compressive strength has a downward trend, it is still higher than the control group. This shows that pretreating the CDWC at 60 °C for 24 h has the best effect.

There may be two reasons for the improvement of CDWGC compressive strength caused by high-temperature pretreating CDWC: First, in a certain range, high temperature improves the reactivity of CDWC [26], making them react more fully in an alkaline environment. Secondly, the higher the pretreatment temperature, the more serious the dewatering of CDWC will be. More water will be absorbed when contacting the alkali solution [27], which leads to higher alkalinity in the matrix, thus increasing the dissolution of aluminosilicate, accelerating the rate of polycondensation geological polymerization reaction, and higher compressive strength of concrete [28]. However, when the alkalinity of the matrix is too high, although hydroxide will make the polycondensation reaction happen faster, it will react prematurely, hindering the hydrolysis of aluminosilicate. Incomplete hydrolysis of aluminosilicate leads to immature molecular structure, and the compressive strength of concrete decreases [29], which also explains why the CDWGC compressive strength starts to decrease when the pretreatment temperature continues to increase.

### 3.2. SEM-EDS and XRD Microanalysis

The micromorphologies of CDWC at different pretreatment temperatures are shown in Figure 5. From the microscopic morphology, it can be found that CDWC contains old aggregates and mortar from waste concrete. After being crushed, the old aggregate presents a blocky shape, while the old mortar presents a fragment shape. The old aggregate is dense, while the old mortar is looser and more porous. However, after different pretreatment temperatures, the micromorphology of CDWC did not significantly change. Compared with CDWC at 20 °C, there are no cracks and holes. This indicates that high-temperature pretreatment of CDWC does not affect its microscopic physical morphology.

EDS results of CDWC with different pretreatment temperatures are shown in Table 6. After pretreatment at different temperatures, the chemical compositions of CDWC have almost no change, the maximum variance of different elements is only 0.3, and the average variance is only 0.09. This shows that high temperature does not damage the chemical composition of CDWC.

The XRD results of CDWC at different pretreatment temperatures are shown in Figure 6. It can be found that after high-temperature pretreatment, the main phases of CDWC are almost unchanged, mainly quartz, calcite, zeolite and mullite. Their main chemical components are silicon dioxide, calcium carbonate and aluminosilicate. High temperature does not destroy the phase of CDWC to form a new phase, which indicates that CDWC can undergo high-temperature curing.

The results of SEM-EDS and XRD both illustrate that high-temperature pretreatment did not change the properties of CDWC. This also confirms the macro mechanical performance conclusion: the CDWC has no negative impact on the compressive strength development stability of CDWGC after experiencing high temperatures.

The reason for strength shrinkage can be explained from two aspects: Firstly, from the development trend of compressive strength under heat curing (Figure 3), the 3-days strength can reach 85.1–93.1% of the 28-day strength, indicating the early strength of CDWGC develops very rapidly under heat curing. When the concrete strength increases to the maximum, the data fluctuation caused by measurement cannot be ignored. Moreover, geopolymer concrete is a brittle material. When measuring brittle material’s mechanical properties, the values’ fluctuation must be considered [22]. On the other hand, the moisture in the concrete will evaporate during the curing process and cause cracks and drying shrinkage of concrete [30], which makes the compressive strength decline.

What’s more, heating curing accelerates the evaporation of water, so strength shrinkage occurs before 28 days. However, after high-temperature pretreatment, the moisture content of CDWC decreases, and the impact of moisture evaporation during the curing process on strength diminishes. Therefore, high-temperature pretreatment CDWC can effectively solve the problem of strength shrinkage.

### 3.3. Effect of Addition CaO on CDWGP Compressive Strength

The influence of different CaO content on CDWGP compressive strength is shown in Figure 7. It can be seen from the research results that the compressive strength of CDWGP increases by 61.8%, 178.4%, and 150%, respectively, when the CaO content is 1%, 3%, and 5%. When the CaO proportion is 0–3%, CDWGP compressive strength increases with the increase of CaO content. When the content exceeds 3%, the compressive strength starts to decline. After adding CaO, the strength of CDWGP may be improved for the following two reasons: First, CaO can be dissolved in the alkaline solution to generate Ca^2+^, which is more active than Na+, and it is easier to combine with Si-O-Si and Al-O-Si to generate C-S-H and C-A-S-H gel [6,31]. This gel is denser than N-A-S-H gel generated by geopolymerization, and its strength is higher. Second, Ca^2+^ can be used as a charge balance ion to accelerate the formation of C-A-S-H and N-A-S-H gel, thus improving its strength [32,33]. However, when the CaO content is too high, it will quickly dissolve in the alkaline solution to form Ca^2+^. When the concentration of Ca^2+^ in the solution exceeds the dissolution equilibrium concentration, high consistency Ca(OH)_2_ will be precipitated in the solution and wrapped on the surface of fly ash and CDWC to prevent further dissolution, thus inhibiting the occurrence of geopolymerization and reducing the strength of the matrix.

According to the previous research results of Dai [17] et al.: heat curing can improve the compressive strength of CDWGP by 86% at 28 days at most. However, this study shows that when the content of calcium oxide is 3%, the compressive strength can be increased by 180.9% under ambient curing conditions, which is much higher than that of heat curing. Therefore, adding CaO effectively improves the compressive strength of CDWGP cured at ambient temperature.

### 3.4. Impact of Improvement Measures on CO_2_ Emissions

This research calculated the CO_2_ emissions using the above strength improvement measures to prepare 1 m^3^ CDWGC and 1 m^3^ CDWGP. At the same time, to more directly reflect the impact of CO_2_ emissions caused by the strength improvement method, this paper introduces the parameter P, which represents the relative proportion of the additional CO_2_ emissions generated by the improvement measures to the total CO_2_ emissions of the concrete prepared without the improvement measures. The CO_2_ emission calculation results are shown in Table 7.

According to the calculation of CO_2_ emissions by LCA, the high-temperature pretreatment CDWC mainly increases the CO_2_ emissions in the concrete preparation stage. At the same time, CaO is an additive material which mainly produces additional CO_2_ emissions in the raw material production and transportation stages. Although the CO_2_ emission increased by high-temperature pretreatment is similar to that of adding CaO, the *p* value is higher. This is due to the higher CDWC and FA used in the paste than in the concrete. The CaO added is relatively more, and the additional CO_2_ emissions are also increased. However, the alkali activator used in the paste is also more, and the total CO_2_ is also significantly increased, so the *p* value of adding CaO is relatively lower. In the two strength improvement methods, adding CaO is more environmentally friendly.

The CO_2_ emissions of CDWGP prepared by heat curing and adding CaO are also compared. When the maximum CaO content is 5%, the CO_2_ emission increases by 105 kg compared with that without CaO, while when the optimal CaO content is 3%, the CO_2_ emission increases only by 63 kg, which is far less than the CO_2_ emission increment of heat curing (203 kg). Kawai [34] et al. calculated the CO_2_ emissions of concrete prepared by different curing methods using LCA and found that heating curing can emit 10 kg more CO_2_ per hour than normal temperature curing. Converted to 24 h of heat curing, the CO_2_ emission increased by about 240 kg, almost the same as the CO_2_ emission value calculated in this work for heat curing. To sum up, the impact of CaO addition on CO_2_ emissions in CDWGP production is far less than that of heat curing. Considering the mechanical properties and CO_2_ emissions, adding CaO can not only effectively improve the compressive strength of CDWGP cured at ambient temperature but also will not cause adverse effects on the environment. Moreover, the effect of this method on improving the compressive strength is better than that of heat curing, and the environmental impact is also far lower than that of heat curing. Therefore, adding CaO into CDWGP is a more reasonable measure to improve the compressive strength in ambient curing conditions.

## 4. Conclusions

This study solved two mechanical imperfections of CDWC-based geopolymer materials through high-temperature pretreating CDWC and adding CaO. Based on the analysis of the data obtained from the test, the following conclusions can be obtained:High-temperature pretreatment CDWC can effectively improve the compressive strength of CDWGC by improving the reactivity of CDWC and increasing the alkalinity of the geopolymer matrix.From the SEM-EDS and XRD results of CDWC, there is no apparent change in the microscopic morphology, and the chemical compositions have few changes after different high-temperature treatments. These indicate the strength shrinkage of CDWGC is not due to the high temperature that destroyed the CDWC and affected its strength development. Instead, high-temperature pretreatment reduces the moisture content of CDWC and cracks caused by the evaporation of moisture inside the concrete during heat curing, thereby improving the stability of concrete strength growth.The mechanical properties of CDWGP cured at ambient conditions can be significantly improved by adding CaO due to the formation of a more compact C-S-H gel and the acceleration of the formation of N-A-S-H gel. But the compressive strength first increases and then decreases with the increase of CaO content, and the optimum amount of CaO is 3%.Compared with the high-temperature pretreatment of CDWC and heat curing CDWGP, the *p* value of adding CaO is smaller, and adding CaO has a better effect on strength improvement. Therefore, adding CaO is a more low-carbon and environmentally friendly way to improve the compressive strength of CDWGP.

## Figures and Tables

**Figure 1 polymers-15-01699-f001:**
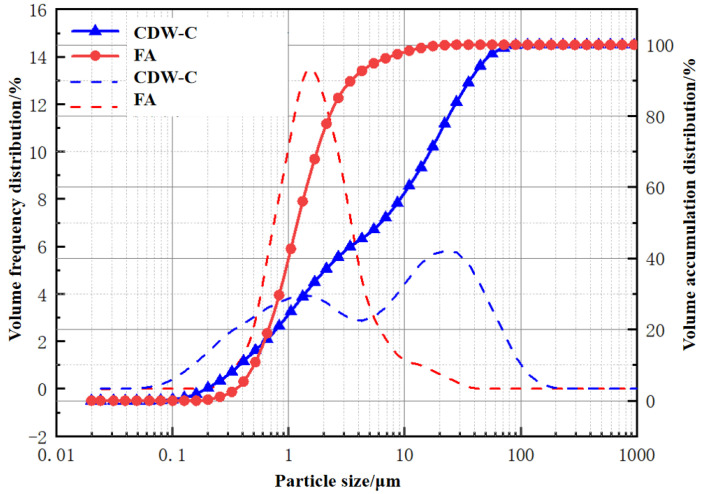
The particle size distribution of FA and CDWC.

**Figure 2 polymers-15-01699-f002:**
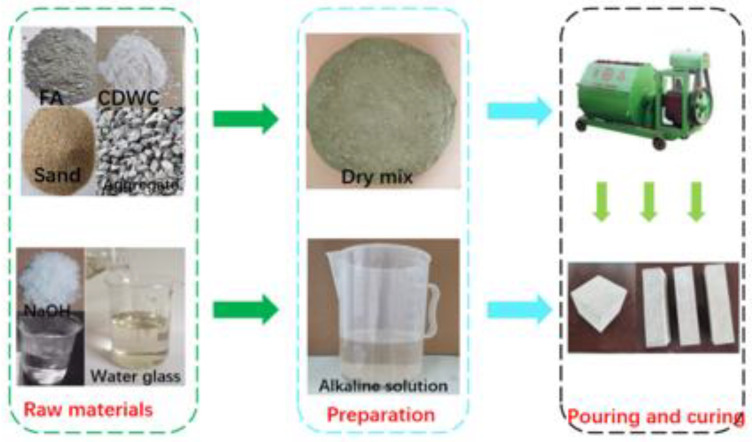
Schematic diagram of specimen preparation.

**Figure 3 polymers-15-01699-f003:**
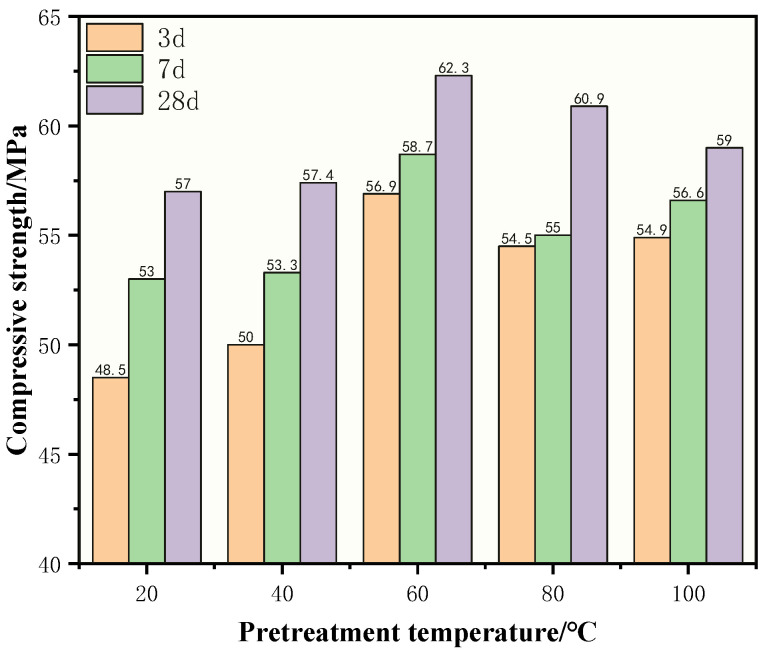
Effect of pretreatment temperature of CDWC on compressive strength development of CDWGC.

**Figure 4 polymers-15-01699-f004:**
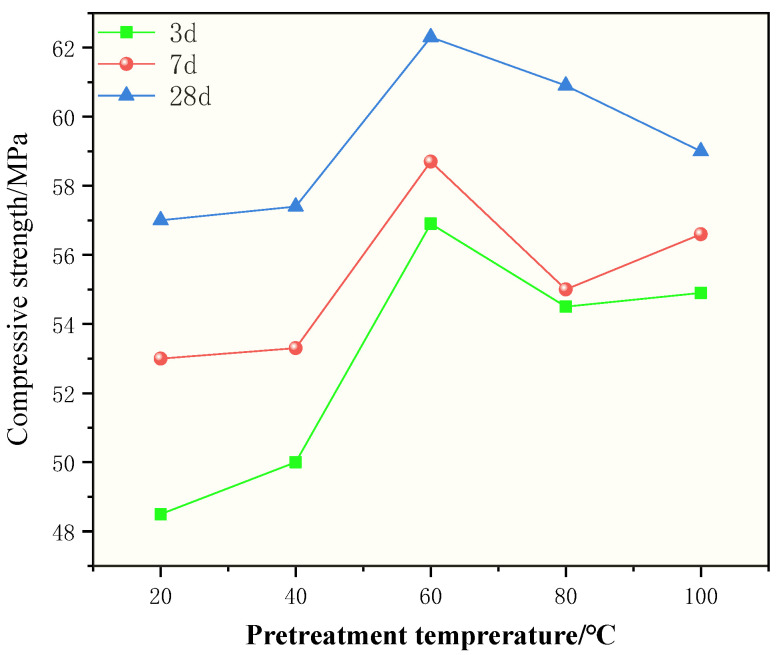
Effect of pretreatment temperature of CDWC on compressive strength of CDWGC.

**Figure 5 polymers-15-01699-f005:**
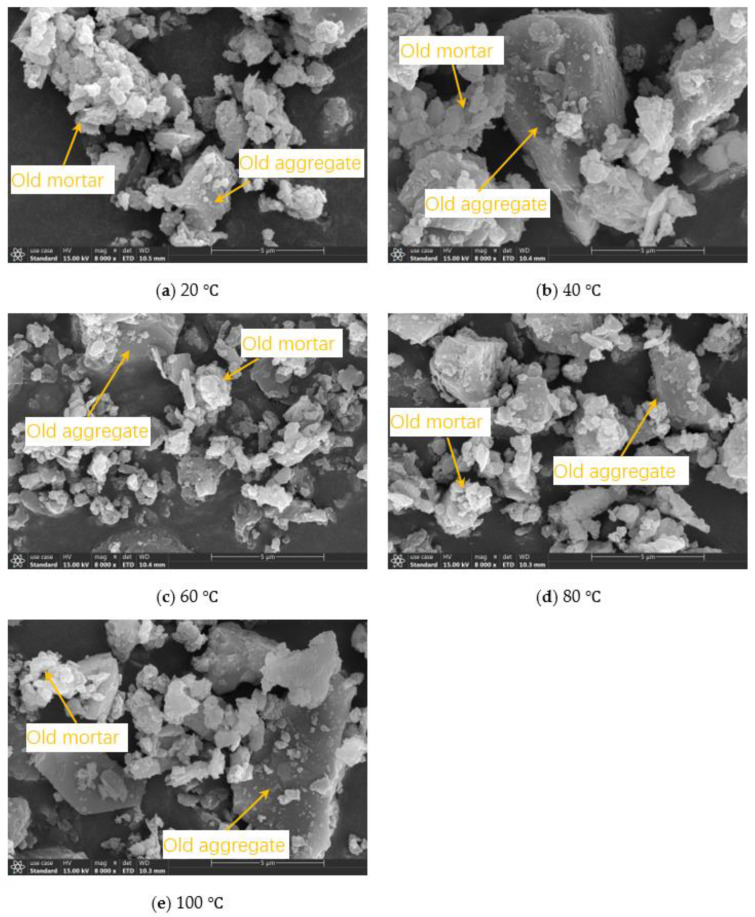
Micromorphology of CDWC after different pretreatment temperatures.

**Figure 6 polymers-15-01699-f006:**
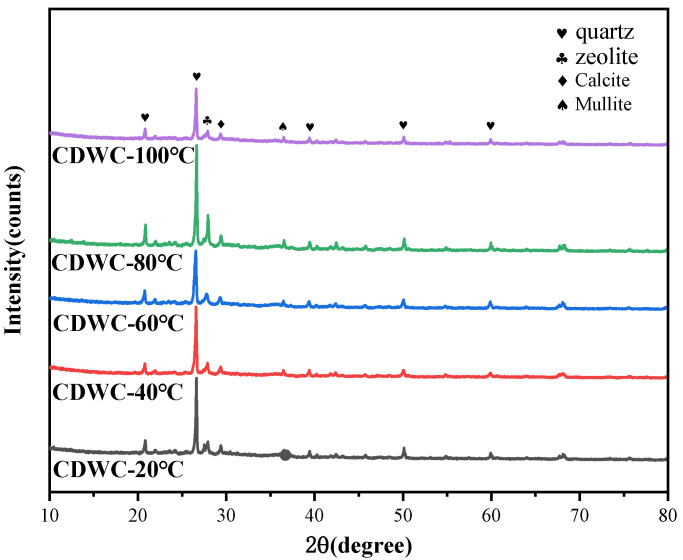
XRD results of CDWC after different pretreatment temperatures.

**Figure 7 polymers-15-01699-f007:**
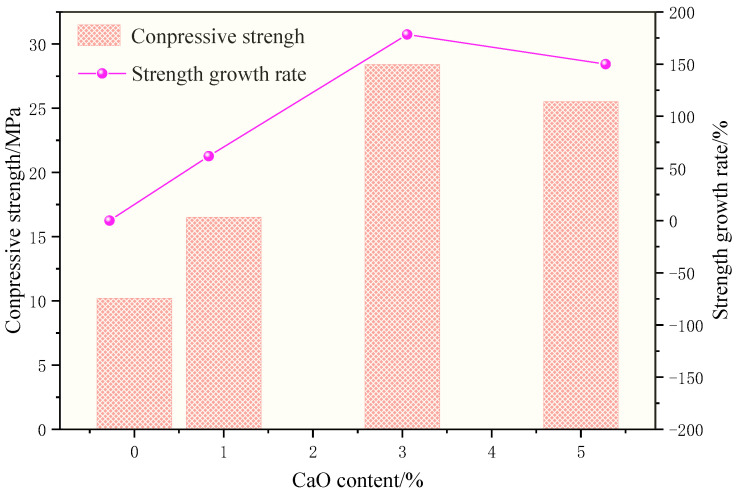
Effect of CaO content on compressive strength of CDWGP.

**Table 1 polymers-15-01699-t001:** Abbreviation list.

Name	Abbreviation
Construction and demolition waste composites	CDWC
Fly ash	FA
CDW-C-based geopolymer concrete	CDWGC
CDW-C-based geopolymer paste	CDWGP
Life cycle assessment	LCA
Scanning electron microscope	SEM
Energy dispersive spectrometer	EDS
X-ray diffraction	XRD
X-ray fluorescence spectroscopy	XRF
Carbon emission ratio parameter	P

**Table 2 polymers-15-01699-t002:** Chemical composition of FA and CDWC.

Composition	SiO_2_	Al_2_O_3_	Fe_2_O_3_	CaO	Na_2_O	P_2_O_5_	TiO_2_	MgO	L.O.I
FA	60.66	14.51	6.29	5.11	0.94	3.42	1.04	-	4.98
CDWC	42.90	10.95	6.35	9.29	0.87	-	-	1.36	26.52

**Table 3 polymers-15-01699-t003:** Physical properties of CDWC.

Physical Properties	Value	Standard Deviations
Average particle size	12.582 μm	1.934
Density	2.60 g/cm^3^	0.101
Water content ratio	<1%	0.0013
Water demand ratio	98.9%	0.009
Liquidity ratio	93.0%	0.039
Strength activity index	73.7%	0.061

**Table 4 polymers-15-01699-t004:** Mix proportion of CDWGC.

Type	Pretreat Temperature °C	FA kg/m^3^	CDWC kg/m^3^	NaOH Malority mol/L	Coarse Aggregate kg/m^3^	Fine Aggregate kg/m^3^	Sodium Silicate kg/m^3^	NaOH kg/m^3^	Water kg/m^3^
CDWGC20	20	368	92	12	1200	540	133.4	21.63	45.07
CDWGC40	40	368	92	12	1200	540	133.4	21.63	45.07
CDWGC60	60	368	92	12	1200	540	133.4	21.63	45.07
CDWGC80	80	368	92	12	1200	540	133.4	21.63	45.07
CDWGC100	100	368	92	12	1200	540	133.4	21.63	45.07

**Table 5 polymers-15-01699-t005:** Mix proportion of CDWGP.

Type	FA kg/m^3^	CDWC kg/m^3^	NaOH Malority mol/L	Sodium Silicate kg/m^3^	NaOH kg/m^3^	CaO kg/m^3^	Water kg/m^3^
CDWGP0	1226.48	306.62	12	444.60	72.09	0	150.21
CDWGP1	1226.48	306.62	12	444.60	72.09	15.33	150.21
CDWGP3	1226.48	306.62	12	444.60	72.09	45.99	150.21
CDWGP5	1226.48	306.62	12	444.60	72. 09	76.66	150.21

**Table 6 polymers-15-01699-t006:** Chemical composition of CDWC at different pretreatment temperatures.

Temperatures (°C)	Mass Ratio of Different Elements (wt%)									
	C	O	Na	Mg	Al	Si	K	Ca	Fe	Total
20	10.30	43.15	0.69	1.03	6.08	20.47	1.55	10.96	5.77	100
40	10.51	43.99	0.67	0.88	5.8	20.65	1.52	10.67	5.02	99.71
60	9.85	43.45	0.98	1.04	6.16	20.51	1.77	10.82	5.42	100
80	10.87	42.46	0.91	0.94	6.72	19.90	1.60	10.45	6.14	100
100	10.39	43.86	0.71	1.09	6.24	19.86	1.63	10.69	5.10	99.59
variance	0.11	0.30	0.02	0.01	0.09	0.11	0.01	0.03	0.18	-

**Table 7 polymers-15-01699-t007:** CO_2_ emissions of producing 1 m^3^ CDWGC or CDWGP (kg CO_2eq_).

Type	Raw Materials Production	Raw Materials Transportation	Concrete Preparation	Total CO_2_ Emissions	Additional CO_2_ Emissions	P (%)
CDWGC20 (control)	319.04	28.72	206.84	554.60	0	0
CDWGC40	319.04	28.72	253.09	600.85	46.25	8.34
CDWGC60	319.04	28.72	260.32	608.08	53.48	9.64
CDWGC80	319.04	28.72	271.68	619.44	64.84	11.69
CDWGC100	319.04	28.72	289.06	636.82	82.22	14.83
CDWGP0 (control)	1048.47	76.24	3.8	1128.51	0	0
CDWGP1	1069.17	76.51	3.8	1149.48	20.97	1.86
CDWGP3	1110.57	77.04	3.8	1191.41	62.9	5.57
CDWGP5	1151.96	77.58	3.8	1233.34	104.51	9.26
CDWGP0 (heat)	1151.96	77.58	3.8	1233.34	206.84	18.33

## Data Availability

Data are contained within the article are available on request from the corresponding author.

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
