# Peer review of "Study on Improving Measures of Mechanical Properties of Geopolymer Materials and Its Effect on CO2 Emission"

_polymers, 2023, doi:10.3390/polym15071699_

Round 1
Reviewer 1 Report
1. Line 16: Instead of "defects", should be replaced with "imperfections". Defect refers to a defined error in the structure of the material, such as line defects (dislocations), point defects (impurities), etc. Also replace "defects" with "imperfections" in lines: 60, 75, 80 and 343.
2. It is generally known that the optimal pretreatment of geopolymer is at temperatures between 60°C and 70°C within 24 hours. Your results also showed that, it's nothing new.
3. Figure 5. At a magnification of 2000 x, you can see almost nothing. Individual segments should be shown at higher magnifications and their detailed analysis should be given.
4. Please compare the chemical composition from Table 1. and Table 5. In Table 1. for CDW-C, there is no data on the carbon content, while in Table 5. for the same material, a carbon content of about 10% appears, which is a rather large amount of carbon. How do you explain that?
Author Response
Dear Reviewer:
Thank you very much for your important comments on our manuscript entitled “Study on improving measures of mechanical properties of geopolymer materials and its effect on CO2 emission” (polymers-2260180). We have revised the manuscript according to your kind advices.
- Line 16: Instead of "defects", should be replaced with "imperfections". Defect refers to a defined error in the structure of the material, such as line defects (dislocations), point defects (impurities), etc. Also replace "defects" with "imperfections" in lines: 60, 75, 80 and 343.
Response 1: Thanks for the reviewer’s comment. We have replaced "defects" with "imperfections" in lines: 16,60, 75, 80 and 343.
- It is generally known that the optimal pretreatment of geopolymer is at temperatures between 60°C and 70°C within 24 hours. Your results also showed that, it's nothing new.
Response 2: Thanks for the reviewer’s comment. Some previous experiments have found that pretreatment at 60-70 ℃ can improve the strength of geopolymer, but our experiments focus more on exploring whether CDWC will be damaged after high temperature, thereby affecting the stability of the strength development of geopolymer concrete. The effective improvement of the strength of geopolymer concrete after pretreatment with CDWC at 60 ℃ is an additional finding of our experiment
- Figure 5. At a magnification of 2000 x, you can see almost nothing. Individual segments should be shown at higher magnifications and their detailed analysis should be given.
Response 3: Thanks for the reviewer’s comment. After pretreating CDWC at different temperatures for 24 hours, we conducted SEM testing again at a magnification of 8000 x and analyzed its microscopic morphology in Part 3.2.
- Please compare the chemical composition from Table 1. and Table 5. In Table 1. for CDW-C, there is no data on the carbon content, while in Table 5. for the same material, a carbon content of about 10% appears, which is a rather large amount of carbon. How do you explain that? .
Response 4: Thanks for the reviewer’s comment. The chemical composition in Table 1 includes the loss on ignition (L.O.I). When measuring the L.O.I, CDW-C has undergone a high temperature of 1000℃. Carbonate and organic matter have been decomposed, so there is no carbon. The chemical composition in Table 5 is obtained by EDS. CDW-C does not undergo a high temperature of 1000℃. Carbonate and carbon-containing organic matter do not decompose, and element C can be detected.

Reviewer 2 Report
Comments and suggestions are in an attachment file.

Author Response
Dear Reviewers:
Thank you very much for your important comments on our manuscript entitled “Study on improving measures of mechanical properties of geopolymer materials and its effect on CO2 emission” (polymers-2260180). We have revised the manuscript according to your kind advices. Please see the attachment

Round 2
Reviewer 2 Report
1. Authors characterized prepared materials only with SEM-EDS. However, for example FTIR and XRD analysis would give more detailed information about the prepared materials. In my opinion at least one of them should be presented.
Response 15: Thanks for the reviewer’s comment. We have subjected CDWC to high-temperature pretreatment for 24 hours, and analyzed the phase composition of CDWC after pretreatment at different temperatures using XRD, and added it to Part 3.2.
Comment: What was temperature? More detailed description of the XRD analysis should be added to the experimental section.
Author Response
Dear Reviewer:
Thanks for your important comments on our manuscript entitled ‘Study on improving measures of mechanical properties of geopolymer materials and its effect on CO2 emission’ (polymers-2260180). The authors appreciate the constructive comments provided by the reviewers. Those comments have been constructive for the authors to improve this manuscript. Now we reply to the comments given as follows:
1. Authors characterized prepared materials only with SEM-EDS. However, for example FTIR and XRD analysis would give more detailed information about the prepared materials. In my opinion at least one of them should be presented.
Response 15: Thanks for the reviewer’s comment. We have subjected CDWC to high-temperature pretreatment for 24 hours, and analyzed the phase composition of CDWC after pretreatment at different temperatures using XRD, and added it to Part 3.2.
Comment: What was temperature? More detailed description of the XRD analysis should be added to the experimental section.
Response : Thanks for the reviewer’s comment. The phase composition characterized by XRD is CDWC pretreated at 20 ℃, 40 ℃, 60 ℃, 80 ℃, and 100 ℃, and we have added the detailed process XRD analysis to the experimental section.
